Designing AI-powered translation education tools: a framework for parallel sentence generation using SauLTC and LLMs

Aleedy Moneerh 1 2
http://orcid.org/0000-0001-7664-9218 Alshihri Fatma 3 faalshehri@pnu.edu.sa
Meshoul Souham 1
Al-Harthi Maha 4
Alramlawi Salwa 5
Aldaihani Badr 6
Shaiba Hadil 5
http://orcid.org/0000-0001-9395-3764 Atwell Eric 2
1 Department of Information Technology, College of Computer and Information Sciences, Princess Nourah bint Abdulrahman University , Riyadh , Saudi Arabia
2 School of Computer Science, University of Leeds , Leeds , United Kingdom
3 Department of Translation, College of Languages, Princess Nourah bint Abdulrahman University , Riyadh , Saudi Arabia
4 Department of Applied Linguistics, College of Languages, Princess Nourah bint Abdulrahman University , Riyadh , Saudi Arabia
5 Department of Computer Science, College of Computer and Information Sciences, Princess Nourah bint Abdulrahman University , Riyadh , Saudi Arabia
6 Princess Nourah bint Abdulrahman University , Riyadh , Saudi Arabia
Alatas Bilal
Electronic publication date: 2025 Mar 31
Publication date: 2025
Volume: 11
Electronic Location ID: e2788
Received 2024 Oct 1; Accepted 2025 Mar 6
Copyright: © 2025 Aleedy et al.
Copyright year: 2025
Copyright holder: Aleedy et al.
License: This is an open access article distributed under the terms of the Creative Commons Attribution License, which permits unrestricted use, distribution, reproduction and adaptation in any medium and for any purpose provided that it is properly attributed. For attribution, the original author(s), title, publication source (PeerJ Computer Science) and either DOI or URL of the article must be cited.
License URL: https://creativecommons.org/licenses/by/4.0/

Keywords: Didactic corpus, Corpus annotation, AI-powered translation education, AI-based translation technology

Funding: Princess Nourah bint Abdulrahman University PNURSP2025R135 This research is funded by Princess Nourah bint Abdulrahman University Researchers Supporting Project number (PNURSP2025R135), Princess Nourah bint Abdulrahman University, Riyadh, Saudi Arabia. The funders had no role in study design, data collection and analysis, decision to publish, or preparation of the manuscript.

==============================
Translation education (TE) demands significant effort from educators due to its labor-intensive nature. Developing computational tools powered by artificial intelligence (AI) can alleviate this burden by automating repetitive tasks, allowing instructors to focus on higher-level pedagogical aspects of translation. This integration of AI has the potential to significantly enhance the efficiency and effectiveness of translation education. The development of effective AI-based tools for TE is hampered by a lack of high-quality, comprehensive datasets tailored to this specific need, especially for Arabic. While the Saudi Learner Translation Corpus (SauLTC), a unidirectional English-to-Arabic parallel corpus, constitutes a valuable resource, its current format is inadequate for generating the parallel sentences required for a didactic translation corpus. This article proposes leveraging large language models like the Generative Pre-trained Transformer (GPT) to transform SauLTC into a parallel sentence corpus. Using cosine similarity and human evaluation, we assessed the quality of the generated parallel sentences, achieving promising results with an 85.2% similarity score using Language-agnostic BERT Sentence Embedding (LaBSE) in conjunction with GPT, outperforming other investigated embedding models. The results demonstrate the potential of AI to address critical dataset challenges in quest of effective data driven solutions to support translation education.

Introduction

As the world becomes increasingly interconnected, often described as a “global village”, translation emerges as a pivotal practice facilitating communication and understanding across diverse languages. Translation bridges cultural differences and plays a vital role in fostering social and cultural development (Zhao, Li & Tian, 2020). The domain of translation education (TE) is central to preparing future translators and interpreters, equipping them with the necessary skills to proficiently translate written and spoken content across linguistic systems. This critical field shoulders the responsibility of enabling global communication. However, TE is inherently dynamic and continuously evolving to keep pace with advancements in translation practices and pedagogical approaches (Igualada & Echeverri, 2019).

Academic translation programs are shaped by two primary forces: the evolving demands of the translation market, driven by technological innovations, and progressive trends in higher education teaching methodologies (Sawyer, Austermühl & Raído, 2019). Recent technological advancements, such as artificial intelligence (AI), big data, and deep learning, have revolutionized the translation industry (Zhao, Li & Tian, 2020; Al-Batineh & Al Tenaijy, 2024). Concurrently, higher education has experienced transformative shifts, including the shift from teacher-centered to student-centered learning models in the late 1990s (Igualada & Echeverri, 2019) and the widespread adoption of competency-based training at the start of the 21st century (Albir, 2017).

To remain relevant, translation programs must adapt their curricula to integrate these technological advancements and align with modern pedagogical practices. This requires the incorporation of state-of-the-art translation tools and the adoption of learning approaches that reflect market needs (Sawyer, Austermühl & Raído, 2019). In response, scholars have underscored the importance of improving existing teaching tools and developing new resources to enhance TE (Zapata, 2016). Key recommendations include revising curricular designs and refining assessment methods to better reflect real-world translation practices (Albir, 2017; He, 2021).

The PACTE group believes that a pivotal aspect of this transformation is the emphasis on translation competence, grounded in cognitive-constructivist and socio-constructivist learning theories (Albir, 2017). The authors in He (2021) advocates for student-led teaching approaches that leverage technological resources. By encouraging students to independently engage with multiple platforms, revise translations, and conduct peer assessments, TE can better reflect the demands of the contemporary translation landscape. Considering emerging technologies, He (2021) emphasizes that translators’ primary role will increasingly involve post-editing machine-generated translations, highlighting the need to cultivate post-editing competencies within TE programs.

The advent of AI, particularly in the domains of natural language processing (NLP) and natural language understanding (NLU), has introduced sophisticated tools capable of supporting these objectives. AI-powered applications can personalize learning experiences, analyze student translations, and automate certain assessment processes, thereby enhancing efficiency and objectivity. Such innovations ensure that students remain competitive in the evolving translation market by acquiring the necessary skills to collaborate with AI tools rather than being replaced by them.

Despite these advancements, a significant challenge persists: the scarcity of high-quality, comprehensive datasets tailored to TE. A parallel corpus, in this context, is defined as a collection of texts in two languages that are direct translations of each other. The Saudi Learner Translation Corpus (SauLTC) represents a notable attempt to address this gap. Developed to facilitate the teaching of translation from English to Arabic (Al-Harthi & Al-Saif, 2019), SauLTC is a multi-version, English-Arabic parallel corpus featuring part-of-speech tagging. It provides two target versions of 366 source texts (STs), allowing for the analysis of translation and revision processes from initial to final drafts. This corpus supports error identification, the evaluation of teaching feedback, and the investigation of individual differences in manual verification. The SauLTC project further aims to develop AI-based translation technologies and eLearning resources to support digital TE.

In alignment with the evolving landscape of TE, this research seeks to build upon SauLTC by developing a parallel didactic corpus enriched with an innovative annotation scheme. A key component of this process is parallel sentence generation, which refers to the automated process of identifying or constructing sentence pairs that convey the same meaning across bilingual texts. Additionally, alignment is the process of matching sentences from two different languages, where each sentence is a translation of the other. This scheme aims to enhance students’ translation competence, foster self-directed learning, and promote autonomy (F Alshihri, M Alharthi, 2024, unpublished data). By focusing on linguistic equivalence between English and Arabic, the annotation highlights structural and lexical correspondences, encouraging learners to reflect on key aspects of the translation process, such as translation units, problem-solving strategies, and the impact of errors.

Moreover, the proposed tool incorporates error analysis and post-editing features, allowing students to identify and rectify inaccuracies systematically. This hands-on approach equips students with the practical skills needed to address real-world translation challenges, bridging the gap between theoretical knowledge and applied practice.

The first stage of this project outlines a framework for transforming SauLTC’s parallel texts into parallel sentences, laying the foundation for future annotations that stimulate critical thinking and active engagement with translation problems. By harnessing AI and large language models (LLMs), the framework facilitates the generation of high-quality parallel sentences, ensuring contextual appropriateness and linguistic precision. The integration of AI-driven tools enhances the pedagogical value of the corpus, offering students a more comprehensive learning experience.

Ultimately, the development of a SauLTC-based parallel sentence dataset promises to advance AI-powered TE tools, providing students with invaluable resources for practical analysis, comparative evaluation, and cultural competency. This initiative supports instructors in delivering targeted feedback, fostering error correction, and enhancing classroom engagement.

The remainder of this article is organized as follows: “Translation and Technology” reviews the role of technology in translation education, while “Related Work” explores recent initiatives in corpus development for TE. “Methodology and Proposed Parallel Sentence Generation Framework” describes the methodology and framework for parallel sentence generation. “Experiment and Results” presents the results and analysis, and “Discussion” concludes with future research directions.

Translation and technology

The rapid advancement of information technology, notably AI technology, is one of the realities affecting translation services and teaching/training (Garbovskiy & Kostikova, 2020; Kanglang & Afzaal, 2021; Wang, 2023; Mohamed et al., 2024; Mohsen, 2024), resulting in a process of constant evolution in the field (Sawyer, Austermühl & Raído, 2019; Kenny, 2019). The translation industry is now AI-driven which improved the levels of machine translation accuracy and efficiency (Zaghlool & Khasawneh, 2024). Machine translation tools and CAT tools are constantly updated and upgraded because of the advances in artificial intelligence technologies (Steigerwald et al., 2022; Mohamed et al., 2024), and this means that the translator, who is expected to translate a huge number of texts quickly in a little amount of time (Rodríguez De Céspedes, 2019), needs to be kept abreast of the latest translation technologies, and therefore, these technologies should be a core component in any TE (Kenny, 2019). In fact, translators will not have a place in today’s translation market if they do not know how to use translation technology (Bowker, 2023).

Technological innovations have led to huge transformations in the translation profession, education, and assessment (Kong, 2022). One of the most prominent transformations is the Computer-Assisted translation (CAT) change of the translators’ role into post-editors (Igualada & Echeverri, 2019). Therefore, translation students need to be trained to use the most recent AI-based translation technologies to enhance their learning experience and empower them to meet the rising demands of the market (Zaghlool & Khasawneh, 2024). Embracing AI technologies provides teachers with ample opportunities to create high-quality translation environments with favorable conditions to improve students’ foreign language skills, improve their translation skills and broaden their horizons, and support active learning (Kong, 2022; Liu, 2022; Wang, 2023).

Related work

Learner translation corpora (LTC) are comprised of translations produced by students, either into their first language (L1) or a second language (L2), aligned with their STs. This subset represents a specialized area within corpus-based translation studies (CBTS), which typically focuses on professional or expert translation corpora (Lefer, 2020). The emergence of LTC began in the early 2000s, approximately a decade following the establishment of CBTS. Foundational initiatives such as PELCRA (Uzar & Waliński, 2001) and MeLLANGE (Castagnoli et al., 2011) were instrumental in shaping and advancing LTC. Espunya (2014) highlights the primary objectives of LTC, which include analyzing the development of translation skills, assessing the efficacy of pedagogical approaches, and developing resources for translator training. Similarly, Andrey & Kunilovskaya (2014) present a comprehensive research plan based on their work with the Russian Learner Translator Corpus (RusLTC), focusing on translation variability, decision-making processes, and the identification of challenging linguistic areas.

Drawing from the MISTiC corpus, Castagnoli (2020) observes that full lexical consistency is most evident in the translation of concrete nouns, functional words, and numerical data, while abstract nouns and metaphorical expressions frequently exhibit greater variability. A more recent contribution to the field of language and translation education is the Multilingual Student Translation Corpus (MUST) which is a multilingual LTC project that provides extensive metadata for translation research (Granger & Lefer, 2020a, 2020b).

Moreover, as part of the MUST initiative, the translation-oriented annotation system (TAS) was developed to enhance translator training and facilitate research on translation quality across various language pairs (Granger & Lefer, 2021).

The multifunctionality of LTC is evident in its diverse applications. These corpora are integral to translator education, offering students exposure to parallel sentence structures that deepen their understanding of linguistic patterns and cultural nuances. They also serve as valuable resources for research on translation quality by facilitating comparative studies of various translation strategies. Furthermore, LTC supports the development of machine translation (MT) systems and computational linguistic studies, underscoring its relevance across both academic and professional domains.

Annotation systems employed in LTC play a critical role in enhancing the analytical process of student translations. Commonly used annotation methods include part of speech (POS) tagging, which labels each word with its grammatical category, allowing for the examination of syntactic patterns and learner language use. Parsing, another key method, analyzes sentence structures to identify syntactic dependencies and relational hierarchies. Error annotation, a cornerstone of LTC research, involves marking errors within learner translations to pinpoint areas of improvement, such as grammatical inconsistencies or lexical inaccuracies.

Several taxonomies within LTC vary widely, ranging from simple frameworks comprising five categories to more elaborate schemes encompassing over 50 classifications. While projects such as RusLTC and KOPTE have developed comprehensive taxonomies for English, French, and German pairs, some annotation systems lack sufficient documentation, posing challenges for achieving consistency in error categorization. Nevertheless, error annotation yields valuable insights into translation competence development. For instance, Wurm (2020) leverages KOPTE error data to investigate the empirical dimensions of translation proficiency, analyzing factors such as time spent abroad, media exposure, and temporal changes in student performance. Her findings indicate that intensive training significantly reduces errors, enhances problem-solving efficiency, and accelerates translation speeds, whereas external variables such as international experience exhibit limited influence on the acquisition of translation competence.

An emerging trend within CBTS is the annotation of corpora focusing on student post editing. Researchers, including Kübler, Mestivier & Pecman (2022), have focused on identifying errors in editing, particularly those involving complex noun phrases in specialized discourse. Their taxonomy categorizes errors into three groups; overconfidence in MT where incorrect MT output remains unchanged; underconfidence in MT, where correct MT output is unnecessarily altered; and failure to correct MT, where errors in MT are acknowledged but not adequately corrected. In their corpus, the error type concerning noun phrases is failure to correct MT.

Further advancing this domain, Lefer, Piette & Bodart (2022) introduced the Machine Translation Post-Editing Annotation System (MTPEAS), It consists of seven categories; value adding edits, successful edits, unnecessary edits, incomplete edits, error introducing edits, unsuccessful edits and missing edits. They have effectively integrated this system with the TAS developed by Granger & Lefer (2021). This classification system helps in structuring annotations for student editing datasets, allowing for the identification and clarification of incorrect segments that persist in the final edited texts. This approach enhances the investigation and understanding of errors encountered during student editing tasks.

These initiatives collectively enrich the scope of LTC research, fostering advancements in translation education and contributing to the diversification of corpus-based translation studies. By accommodating multiple languages, text genres, and proficiency levels, LTC continues to serve as a vital resource for advancing translation pedagogy and practice.

SauLTC represents a unidirectional, multi-version parallel LTC, comprising student translators’ graduation projects, some of which have undergone professional editing and subsequent publication. SauLTC offers two distinct translation versions: a sub-corpus of unedited draft translations reflecting the initial outputs of student translators, and a final sub-corpus containing professionally edited versions. The SauLTC software facilitates both independent searches within each sub-corpus and parallel searches across two or three sub-corpora. The availability of draft and final submission sub-corpora allows for the differentiation between linguistic features attributable to the initial translation process and those arising from post-feedback revisions. By analyzing both draft and final versions, researchers can gain deeper insight into the translation process, offering valuable contributions to the understanding of translation development and competence (Kruger, 2012; Bisiada, 2017). This dual-layer structure enables the identification of mediations occurring during the transition from source to target text and through subsequent editorial interventions. These mediations may manifest as “translationese” (unusual features characteristic of translated texts) or as editorial modifications aimed at enhancing lexical diversity, simplification, or explicitation. The dual availability of draft and final versions positions SauLTC as an essential resource for process-oriented, corpus-based translation studies. The authors in Alasmri & Kruger (2018) highlighted the scarcity of corpus-based translation Arabic research. Thus, to address this scarcity, particularly in learner corpus research and corpus-based translation studies, this article works towards a more accurate sentence-alignment system for SauLTC.

Sentence alignment methods fall into three main categories: length-based, lexical-based, and neural approaches. Length-based models (Church et al., 1993) align sentences based on length correlation, which has been effective for some European languages but struggle with morphologically rich languages and those with different character structures. Lexical-based methods (Varga et al., 2007) align sentences using bilingual dictionaries and word-overlap heuristics, making them effective for direct translation equivalents. However, they struggle with structurally divergent translations and paraphrased sentences, where direct word matches are insufficient for accurate alignment. Authors in Grégoire & Langlais (2018) introduced a neural-based parallel sentence extraction system that maps sentences into a shared vector space, improving alignment accuracy over traditional methods. Their fully neural approach eliminates the need for predefined feature engineering but relies on large-scale parallel corpora, making it less applicable to low-resource languages. Additionally, their model does not handle out-of-vocabulary (OOV) words, which may lead to alignment errors when dealing with rare or unseen terms. The Parallel Hierarchical Attention Network (PHAN) (Zhu, Yang & Xu, 2020) further improves alignment by assigning different weights to key words, yet it also depends on large parallel datasets for training, posing challenges for low-resource language pairs.

Methodology and proposed parallel sentence generation framework

To address the problem at hand, this section details the methodological framework employed in this study to derive a didactic corpus of parallel sentences from the SauLTC. The chosen approach ensures a robust and systematic investigation of recent LLMs in achieving parallel sentence generation from document pairs. The following sub-sections dive into the specific methods utilized to achieve the research objectives. Figure 1 shows the end-to-end methodological framework adopted in our study. It encompasses the following main phases:

Figure 1 Proposed methodological framework.

Data collection and understanding

As stated earlier, we use the SauLTC developed as part of the “Design and Compilation of the Saudi Bilingual Corpus for Translation Learners” project (Al-Harthi & Al-Saif, 2019). SauLTC is an English-Arabic translation corpus comprising three sub-corpora and three participant profiles. The first corpus consists of English student STs; the second and third corpora include two versions of the translations into Modern Standard Arabic (MSA).

The SauLTC project involved three types of participants. The first and primary group consisted of 366 senior Saudi female translation students, with a mean age of 21.84 years (standard deviation (SD) = 0.64). These students were in the final stages of their academic program, ensuring a consistent level of translation proficiency for the study. The student profile contains information about the student translator, the source, and the translated text. The second group of participants included 48 women instructors who provided feedback on students’ draft translations and later assessed their final submissions. All instructors hold at least a master’s degree in translation or linguistics and with diverse employment backgrounds. Some instructors were college faculty members, while others were freelancers or loan faculty members from other universities in Saudi Arabia (see Al-Harthi et al., 2024) for more details. Instructor profiles document their educational background, teaching experience, and expertise in supervising translation projects. The third group of participants was composed of 23 alignment verifiers who reviewed and verified the accuracy of the automatic sentence alignments.

The STs used in the SauLTC corpus are chapters or extracts from booklets, averaging approximately 6,000 words each. Students had the option to select their own texts for translation, provided they obtained instructor approval for the book title before beginning the translation process. These texts cover a variety of genres, as detailed in Table 1, including psychology, self-help, business, parenting, language, religion, education, biography, autobiography, nutrition, management, fiction, social sciences, sciences, history, and health-related texts. To address potential overlaps between genres like “business” and “management,” we defined them based on scope and focus. “Business” covers commercial activities, market strategies, and economic interactions, while “management” focuses on organizational processes, leadership, and decision-making. Classification was guided by keyword analysis, content review, and purpose-based categorization. Ambiguous cases were resolved by prioritizing the text’s primary focus, ensuring clear distinctions for data interpretation. Table 1 lists the text genres available in the database (Al-Harthi & Al-Saif, 2019).

Table 1 SauLTC text genres.

Text genres	Number of texts in the database under each text genre	
Health	109	
Psychology	41	
Self-help	66	
Business	51	
Parenting	12	
Language	12	
Religion	1	
Education	28	
Biography	1	
Autobiography	20	
Nutrition	11	
Management	1	
Fiction	7	
Social sciences	1	
Sciences	3	
History	2	
Total	366	

Data selection

This section outlines our approach to selecting the essential parts of the SauLTC for our translation learning system. The dataset was carefully reviewed to identify the most relevant components for our study. We specifically focused on the dataset that contains English texts alongside their corresponding Arabic translations. This targeted selection from the dataset ensures that our analysis is precisely aligned with the needs of bilingual translation studies. Table 2 below shows the description of the related metadata.

Table 2 Dataset metadata.

Column name	Description	
ID	Row sequence number	
Type	Type of translation (SOURCE, DRAFT, FINAL)	
groupNumber	Translation file group number	
transID	Translator ID	
Version	Translation version	
Year	Translation submission year	
Text	The English or Arabic text	
Genre	The classification of the text	

We targeted health-related texts for our analysis as they represent the largest subset of translations in the corpus, comprising 109 texts. This substantial subset allowed for a detailed exploration of their linguistic and translational features while reflecting the predominant composition of the corpus. However, the framework used in this study is designed to work equally well with all text genres, requiring no specific adjustments or handling for different genres. This ensures that the findings are applicable across various genres within the SauLTC and that the alignment process remains consistent regardless of genre.

Data preparation

The data preparation phase is critical in developing LLMs, especially when dealing with the complexities of English-Arabic texts. This phase involves several key steps to ensure the dataset is optimized for subsequent analyses: Initial data assessment and correction: We began with a comprehensive review of the dataset to identify and document any discrepancies or mismatches. This included checking for instances where English STs did not have corresponding Arabic translations and vice versa. We also addressed and corrected any data entry errors, particularly where texts are mislabeled or placed in incorrect database columns. Ensuring that each text is correctly categorized is essential for accurate data handling and subsequent processes.

Data cleaning: We meticulously cleaned the dataset by identifying and eliminating non-essential columns such as ID, group number, version, and year. This step simplifies the dataset and focuses attention on the core aspects of the translation corpus.

Data filtering: We filtered out all entries containing preliminary draft texts, opting to retain only those entries that included the final revised versions of the Arabic translations corresponding to their English STs. This selective inclusion ensures that our analysis is based on the most accurate and refined data available.

Text preprocessing: The text preprocessing phase involved carefully planned steps to preserve the integrity and meaning of the text while addressing formatting irregularities. Regular expressions were used to handle issues such as extraneous spaces, newline characters (\n), line breaks (\\n), and non-breaking spaces (\xa0), ensuring a clean and standardized text format. This approach maintained the original structure and semantics, making the data suitable for alignment and analysis. Figure 2 shows a general example of text preprocessing process.

Figure 2 Preprocessing examples.

Document pairs generation

Once the dataset was cleaned and verified, advanced scripting and database management tools were employed to automate the pairing of English STs with their corresponding Arabic translations. This process ensured the accuracy and efficiency of generating document pairs.

Parallel sentences generation

Following the successful pairing of text documents, the next pivotal step in our methodology involves the generation of parallel sentences pairs in English and Arabic. For this task, we employed the Generative Pre-trained Transformer (GPT) model, developed by OpenAI. GPT leverages transformer-based architecture and serves as a prime example of a LLM. Its versions GPT-3 and GPT-4 are some of the most well-known and powerful LLMs currently available. This model is trained on a diverse corpus of text data and excels in recognizing and generating sentence boundaries, not merely based on punctuation but through a deep contextual understanding of language syntax and structure. GPT’s sophisticated capabilities allow it to handle nuanced expressions and complex grammatical constructions effectively, surpassing traditional methods in segmenting and aligning sentences in complex texts (Wu et al., 2023).

To facilitate sentence alignment using this LLM through GPT, we implemented a multi-step methodology to address some limitations of the GPT model. Initially, we utilized a Sentence Embedding model to segment the texts into smaller units based on semantic similarity and a predefined token count limit for English.

The segmentation process began by splitting the English text into smaller units, each limited to approximately 1,000 tokens. This process employed sentence-level tokenization to segment the text into coherent, semantically meaningful units. Sentence boundaries were identified using basic delimiters, such as punctuation and whitespace, ensuring sentence structure was preserved while adhering to the token limit.

For Arabic, corresponding segments were identified dynamically using multilingual embeddings to ensure semantic equivalence with the English segments. Specifically, the cosine similarity of sentence embeddings was calculated to match the last sentence in each English segment with the most semantically similar Arabic sentences. This dynamic alignment determined the Arabic segment boundaries, ensuring contextual and semantic alignment with the English segments.

Although the Arabic segments were not explicitly constrained by a fixed token limit, the semantic alignment process ensured they were approximately equivalent in length to the English segments in terms of tokens. The underlying Sentence Embedding model internally employs subword-level tokenization, enabling accurate representation of linguistic nuances in both English and Arabic, including morphological variations and structural differences.

By integrating sentence-level segmentation for English and semantic similarity-based alignment for Arabic, this approach ensures that the English and Arabic segments are balanced in terms of tokens and content, making them suitable for subsequent parallel sentence generation.

For sentence alignments, each segment pair is processed by GPT to split the segments into aligned sentences. The following prompt was used to instruct GPT: Prompt: “Segment each English text and Arabic text, then respond by aligning each sentence in the English text with its corresponding Arabic text. Format each pair as ‘English: (English Text) Arabic: (Arabic Text)’. Separate each sentence pair with a newline.”

The responses from GPT are long texts where each English text segment is introduced with the keyword “English:” followed by its corresponding Arabic translation, which starts with the keyword “Arabic:”.

Despite GPT’s advanced capabilities, it may occasionally encounter technical and formatting issues, necessitating a thorough review and correction process to ensure accuracy. In this study, we assess the quality of GPT’s responses and make necessary adjustments to align with our quality criteria. These criteria include ensuring there are no null values, and that all data adheres to the correct format. Responses that meet these standards are archived for future use, while discrepancies are carefully documented for in-depth analysis.

The detailed procedure for generating parallel sentences is outlined in the pseudocode illustrated in Fig. 3.

Figure 3 Pseudocode of the bilingual text segmentation, aligning using GPT.

Evaluation

The evaluation phase of our study plays a critical role in assessing the effectiveness and accuracy of the parallel sentence pairs generated in the previous stages. This phase utilized a dual-method approach: quantitative analysis through advanced multilingual sentence embedding models and qualitative analysis via expert human review.

The quantitative analysis involved transforming sentences into numerical vectors using several multilingual sentence embedding models. This conversion allows for a robust semantic alignment analysis between the English and Arabic sentences by calculating cosine similarity scores. The proposed model in this research includes: the Universal Sentence Encoder (mUSE) (Reimers & Gurevych, 2020; Feng et al., 2020), MiniLM (Kurek et al., 2024), MPNet (Kurek et al., 2024) and the Language-agnostic BERT Sentence Embedding (LaBSE) (Reimers & Gurevych, 2020; Feng et al., 2020). These proposed models were selected for their: Efficacy in handling multilingual content: These models are trained on massive datasets encompassing multiple languages, enabling them to understand the nuances of different languages and produce comparable embeddings.

Ability to capture nuanced linguistic features: By leveraging powerful architectures like transformers, these models go beyond simple word co-occurrence and capture deeper semantic relationships within sentences.

Once the text segments are transformed into embeddings, we employ cosine similarity to measure the semantic similarity between each pair of English and Arabic vectors. Cosine similarity is a metric used to determine how similar two vectors are, irrespective of their size. Mathematically, it measures the cosine of the angle between two vectors projected in a multi-dimensional space. The cosine similarity formula, as described by Alfarizy & Mandala (2022), is given in Eq. (1):

(1) CosineSimilarity=cos(θ)=A.B|A||B|

where A and B are the vector representations of the English and Arabic text segments, respectively. The resulting value ranges from −1 to 1: 1: The texts are exactly similar.

0: The texts have no similarity.

−1: The texts are exactly opposite.

To ensure accurate alignment, we established a cosine similarity threshold of 0.7 for identifying semantically aligned sentence pairs. This threshold was chosen based on empirical findings from studies in multilingual sentence alignment tasks. For instance, Chimoto & Bassett (2022) demonstrated that restricting sentence pairs to those with cosine similarity scores above 0.7 yielded alignment accuracies exceeding 85%.

However, the effectiveness of a 0.7 threshold can vary depending on the embedding model used. As discussed in Song et al. (2024), cosine similarity thresholds are influenced by the training methodologies of embedding models, which may produce higher similarity scores even for dissimilar sentences. To address this, the human evaluation process plays a critical role in identifying and addressing such cases where cosine similarity alone may fail to capture nuanced semantic differences.

In the human evaluation process, a representative subset of 100 sentence pairs was selected for review. Random sampling was employed to minimize selection bias and ensure the sample reflected the broader dataset’s characteristics. Random selection is a widely recognized method in computational linguistics for maintaining generalizability and avoiding overfitting to specific subsets (Saldías et al., 2022). A sample size of 100 pairs balances practical feasibility with statistical reliability, as demonstrated by similar practices in machine translation evaluation studies, such as the 100-sentence test sets used for validation in English Korean translation experiments (Park & Padó, 2024).

The evaluators involved in the human assessment were bilingual experts with extensive experience in translation, linguistic analysis, and corpus linguistics. They were university faculty members specializing in translation education and corpus linguistics, ensuring pedagogical and practical insights into translation quality assessment. Both evaluators had been involved in multiple evaluation processes and underwent a short training phase before commencing evaluation.

Evaluators utilized a standardized rubric inspired by the Multidimensional Quality Metrics (MQM) framework (Freitag et al., 2021; Lommel et al., 2024), and were familiarized with the evaluation criteria, which covered addition, omission, and mispairing errors. The rubric minimizes bias by applying standardized, objective criteria for error identification and scoring, ensuring consistency across evaluations (Freitag et al., 2021; Lommel et al., 2024). Our evaluation focused on three key factors related to accuracy: addition, which refers to extra information added that is not present in the source; omission, which involves information missing that is present in the source; and mispairing, where the incorrect meaning is conveyed compared to the source (Freitag et al., 2021; Lommel et al., 2024).

Each error type was classified into one of three severity levels: minor errors, which do not significantly impact the understanding or usability of the translation; major errors, which affect understanding or usability but can still be inferred; and critical errors, which severely impact understanding or usability and may cause significant misunderstandings. These severity levels were assigned specific weights—minor (1), major (5), and critical (25)—to reflect their impact on translation quality (Freitag et al., 2021; Lommel et al., 2024). This classification aligns with other frameworks in translation quality evaluation. For instance, O’Brien (2012) defines minor errors as those that are noticeable but do not impact usability, major errors as those that negatively affect meaning, and critical errors as those that significantly impair usability, safety, or behaviour. These definitions align closely with those outlined by Malcolm (2004), who emphasizes the varying degrees of impact on functionality and safety.

The error types and severities are illustrated in Table 3, which provides examples for each category to clarify their impact on the accuracy of the sentence pair alignment.

Table 3 Examples of error types, evaluation scales, and their impact on sentence alignment quality.

Error type	Evaluation scale	Source (English)	Target (Arabic)	Explanation	
Addition	Minor	Take a shower and then dry the body.	خذ حمامًا ثم جفف الجسم تمامًا بمنشفة دافئة.	The addition of “بمنشفة دافئة” (“with a warm towel”) introduces extra, non-essential information but does not significantly alter the meaning.	
Major	Platelets are essential for the body’s ability to control bleeding.	الصفائح الدموية ضرورية لقدرة الجسم على التحكم في الدم ومنع تجلطه.	Adding the phrase “ومنع تجلطه” (stop clotting), which gives inaccurate information, is a major error in the translation, negatively affecting meaning and misleading the audience.	
Critical	Do not mix these chemicals together.	لا تخلط هذه المواد الكيميائية بعضها البعض لأنها آمنة للاستخدام.	Adding “لأنها آمنة للاستخدام” (“because they are safe to use”) misrepresents the warning, creating a critical misunderstanding.	
Omission	Minor	The baby is crying loudly.	الطفل يبكي.	The omission of “loudly” (“بصوت عالٍ”) reduces detail but does not affect the primary meaning of the sentence.	
Major	Use disposable needles for one-time use.	استخدام الإبر مرة واحدة فقط.	The omission of “disposable” (“الإبر المخصصة للاستعمال الواحد”) can cause ambiguity, potentially leading to reuse of needles meant for disposal.	
Critical	Examine blood donors for safety and ensure it is free of the AIDS virus.	افحص المتبرعين بالدم.	Omitting “ensure it is free of the AIDS virus” removes vital information about screening for specific, critical risks.	
Mispairing	Minor	Take the pill after meals.	تناول الحبة أثناء الأكل.	The target text “أثناء الأكل” (“during meals”) introduces a slight timing discrepancy compared to the source instruction “after meals,” creating a minor mismatch in meaning.	
Major	For external use only.	للاستخدام في الخارج فقط.	The target text “للاستخدام في الخارج فقط” (“To be used outdoors only”) refers to location where the medication should be used, significantly altering the intended meaning of the ST, which instructs users on how the medication should be used and thus creating a major mismatch in meaning.	
Critical	Consult your doctor if symptoms persist.	استمر في تناول الدواء حتى تختفي الأعراض.	The target text “استمر في تناول الدواء حتى تختفي الأعراض” (“continue taking the medication until symptoms disappear”) replaces the source instruction with a completely unrelated statement, misrepresenting the original message.	

The overall MQM quality score was calculated using Eq. (2):

(2) MQMQualityScore=100−(∑i=1n⁡wi.eiT.100)

where wi represents the weight of each error type, ei is the frequency of errors, and T is the total evaluated sample.

Errors identified during human evaluation were categorized, weighted, and summed according to MQM guidelines to derive the final quality score (Lommel et al., 2024).

To ensure the consistency and reliability of human judgments, Gwet’s AC1 coefficient was computed as a measure of inter-rater reliability. It is recognized for its strengths in providing reliable agreement measures even under challenging rating conditions, such as imbalanced data distributions and small sample sizes (Gwet, 2008; Wongpakaran et al., 2013).

Each evaluator independently assessed the same 100-sentence subset, assigning severity ratings to identified errors. These ratings were then recorded and analyzed to quantify the level of agreement. Gwet’s AC1 was calculated using the probabilistic model (Gwet, 2008), which accounts for the probability of agreement due to chance and adjusts accordingly. The AC1 values were interpreted using the benchmark scale, where values below 0.20 indicate slight agreement, 0.21–0.40 fair agreement, 0.41–0.60 moderate agreement, 0.61–0.80 substantial agreement, and values above 0.81 represent almost perfect agreement (Wongpakaran et al., 2013).

The Gwet’s AC1 Coefficient was calculated using Eq. (3):

(3) AC1=1−(Do−De1−De)

where: Do = Observed disagreement, calculated as the proportion of instances where evaluators provided different ratings.

De = Expected disagreement by chance, determined based on the assumption that ratings are assigned randomly.

In cases where evaluators disagreed significantly, discrepancies were reviewed and discussed to refine judgment criteria. This process helped ensure consistency before final evaluation.

Human evaluators played a crucial role not only in scoring but also in offering qualitative feedback. Their observations highlighted specific alignment challenges and provided deeper context for interpreting results.

Experiment and results

In this section, we detail the experimental setup and procedures used to test and validate the parallel sentence generation framework previously described. This experiment was specifically designed to evaluate the effectiveness of our methodology in generating high-quality English-Arabic sentence pairs with a particular focus on the health text genre within the SauLTC.

Following the completion of the data collection, comprehension, selection, preparation, and document pair generation phases, as elaborated in the preceding section, a total of 85 document pairs were successfully created and prepared for use in the parallel sentence generation phase. These document pairs were subsequently segmented into smaller units to facilitate processing by GPT. The segmentation process involved splitting the English text into smaller units, each limited to approximately 1,000 tokens, using sentence-level tokenization. Arabic segments were dynamically aligned with their corresponding English segments by leveraging semantic similarity scores calculated using the paraphrase-multilingual-mpnet-base-v2 embedding model. This ensured contextual and semantic equivalence between the segments, even though Arabic segments were not constrained by a fixed token limit. The technical details of this process are outlined in Table 4.

Table 4 Technical specifications.

Technical specification	
Embedding model for segment alignment	paraphrase-multilingual-mpnet-base-v2, selected for multilingual capabilities and semantic accuracy in alignment tasks.	
Segmentation methodology	Sentence-based segmentation for English (token limit: 1,000); dynamic token-based alignment for Arabic.	
Tokenization approach	Sentence-level tokenization for segmentation; subword-level tokenization within embeddings.	
GPT model	gpt-3.5-turbo-16k	
Prompt	Segment each English text and Arabic texts, then respond by aligning each sentence in the English text with its corresponding Arabic text. Format each pair as: “English: (English Text) Arabic: (Arabic Text)” Separate each sentence pair with a newline.	
Number of tokens per message	Approximately 2,000 tokens per message (1,000 for English, dynamically equivalent for Arabic).	
Temperature	Set to 0.4 to prioritize deterministic and consistent outputs with minimal randomness in sentence alignment.	
Number of messages to GPT	469 total API calls made to process the dataset.	

The segmented pairs were processed by GPT, configured as described in Table 4, to generate aligned English-Arabic sentence pairs. The experiment utilized the GPT-3.5 Turbo model, chosen for its balance between accessibility, cost-effectiveness, and suitability for the task. The process used a structured prompt to align segments, producing responses with clear formatting. GPT outputs were designed to pair each English segment with its corresponding Arabic translation, marked with the keywords “English:” and “Arabic:” for clarity.

Upon generating the sentence pairs, each response underwent a thorough evaluation to verify its adherence to the quality and formatting standards established for this study. Responses with mismatched English and Arabic counts, improper formatting, or alignment errors were flagged for reprocessing. These flagged outputs were either corrected manually or excluded if they failed to meet the quality criteria upon re-evaluation. During the evaluation process, several issues were identified in the alignment of English-Arabic parallel sentences; examples are presented in Table 5.

Table 5 Examples of the identified issues in English-Arabic parallel sentence alignment.

Problem	Example	
Incomplete	English: Its corresponding Arabic translation is missing.	
Arabic: (Empty)	
Duplication	English: They also just look cool.	
Arabic: كما أنها تبدو رائعة.	
English: They also just look cool.	
Arabic: كما أنها تبدو رائعة.	
Language switching	English: الحقيقة هي أن هذا لا يعمل دائمًا	
Arabic: The truth is that this doesn’t always work.	
Repeated Arabic text	English: !تذكر أن عملية الهضم تحرق الطاقة	
Arabic: تذكر أن عملية الهضم تحرق الطاقة!	
Formatting	Building on Enthusiasms	
English: Building on Enthusiasms	
Arabic: مبني على الحماس	
I spoke a bit about this in the chapter on conception.	
Arabic: تحدثت قليلاً عن هذا في الفصل المتعلق بالحمل.	
October It’s the scariest month of the year, but never fear … we have tips to get you through the candy, cold weather, and other challenges of October.	
Arabic: أكتوبر هو أكثر الشهور رعبًا في السنة، ولكن لا تخف… لدينا نصائح لمساعدتك في التغلب على الحلوى والطقس البارد والتحديات الأخرى التي تواجهك في أكتوبر.	

These examples illustrate the challenges in ensuring accurate alignment of parallel sentences and demonstrate the importance of our review and correction processes.

Of the 469 initial responses generated by GPT model, 443 were accepted as meeting the high standards set for inclusion in the study. The remaining 26 responses did not meet the necessary criteria and were therefore discarded. This process of validation ensures that our data is both reliable and robust, suitable for further analysis and application in translation studies. Ultimately, the results of the sentence alignment processes are organized into a structured table, formatted with two columns—one for English and one for Arabic, as depicted in Fig. 4.

Figure 4 Example of converting GPT responses.

As a result from the sentence generation phase, we generate a total of 15,845 English-Arabic sentence pairs. The aligned sentence pairs were then evaluated through a dual approach—quantitative analysis and expert human review. Quantitative assessment involved the use of four multilingual sentence embedding models to convert the sentences into vectors for semantic alignment analysis. The models selected for this experiment included: “distiluse-base-multilingual-cased-v2”: based on the Universal Sentence Encoder (mUSE).

“paraphrase-multilingual-MiniLM-L12-v2”: a smaller and more efficient version of the MiniLM architecture.

“paraphrase-multilingual-mpnet-base-v2”: based on the MPNet.

“LaBSE”: The Language-agnostic BERT Sentence Embedding (LaBSE).

Following the transformation of sentences into vector embeddings using the selected multilingual models, the semantic alignment between English and Arabic sentence pairs was assessed using cosine similarity, a metric that quantifies semantic similarity based on vector relationships. The average cosine similarity scores and SD for each model are presented in Table 6, reflecting their alignment performance and variability.

Table 6 Average cosine similarity scores and SD.

Embedding models	Average cosine similarity scores	SD	
Distiluse-base-multilingual-cased-v2	0.828	0.11	
Paraphrase-multilingual-MiniLM-L12-v2	0.826	0.12	
Paraphrase-multilingual-mpnet-base-v2	0.863	0.10	
LaBSE	0.852	0.09	

Among the evaluated models, Mpnet achieved the highest average similarity score 0.863, indicating effective semantic alignment. LaBSE followed closely with a score of 0.852 and demonstrated greater consistency, as evidenced by its lower standard deviation (SD = 0.09). These results suggest that while Mpnet provides strong overall alignment, LaBSE offers more consistent performance across diverse sentence pairs.

A cosine similarity threshold of 0.7 was used to categorize sentence pairs: Aligned pairs: Scores ≥ 0.7 were considered semantically aligned.

Misaligned pairs: Scores < 0.7 indicated misalignments.

Figure 5 illustrates the distribution of cosine similarity scores, with the threshold prominently marked.

Figure 5 score_distributions_comparison_uniform_frequency.

At the 0.7 threshold, LaBSE aligned the highest percentage of sentence pairs (95%), followed by Mpnet (94%), Distiluse (91%), and MiniLM (88%). While these percentages provide a broad overview, examining specific examples shows more about each model’s performance.

For instance, the English sentence “Instead, your body will try to keep you alive by slowing down its metabolism” and its Arabic equivalent “بدلاً من ذلك، سيحاول جسمك الحفاظ على حياتك من خلال إبطاء عملية الأيض” were scored differently across models. LaBSE achieved the highest score (0.919), closely followed by Mpnet (0.841), demonstrating their ability to align nuanced sentence pairs effectively. In contrast, Distiluse (0.656) and MiniLM (0.646) struggled to capture the full semantic equivalence, scoring below the 0.7 threshold.

This example aligns with the overall percentages, indicating that LaBSE and Mpnet are more suitable for tasks requiring precise semantic alignment.

After calculating the cosine similarity scores, a comprehensive human evaluation was conducted to further validate the quality of the aligned sentence pairs. A subset of 100 sentence pairs was evaluated by human experts based on predefined criteria outlined in the Methodology section. The human evaluators assigned an average MQM quality score of 85%, indicating a high level of satisfaction with the results. To ensure the consistency and reliability of human judgments, Gwet’s AC1 Coefficient was computed as a measure of inter-rater reliability, yielding a score of 0.98, which indicates almost perfect agreement.

A key observation was that sentence pairs with high cosine similarity scores (>0.70) generally aligned well with human evaluations, confirming the reliability of cosine similarity as an initial filtering metric. However, cases involving idiomatic expressions or culturally adapted translations sometimes received lower similarity scores due to differences in surface-level wording. For example, the English phrase “We want what we are told we can’t have” was translated as “فكل ممنوع مرغوبا”, receiving a similarity score of less than 0.5 across all models. Despite the low score, human evaluators recognized it as a valid translation, illustrating the limitations of cosine similarity in capturing deeper semantic relationships.

Overall, our study confirms that cosine similarity serves as a useful complementary tool in translation evaluation, particularly for identifying lexical alignment. However, human judgment remains essential for assessing meaning in cases involving idiomatic expressions or significant paraphrasing.

Discussion

This study introduces a novel approach to parallel sentence alignment by integrating GPT, advanced sentence embedding models and human expert reviews into a single, adaptive framework.

Traditional length-based and lexical-based methods rely on fixed heuristics or bilingual dictionaries, which struggle with paraphrased translations and structurally divergent sentences. Similarly, neural models improve alignment by mapping sentences into shared vector spaces, but they typically require large parallel datasets for training, making them less suitable for low-resource languages. Many of these models also struggle with out-of-vocabulary words, affecting alignment accuracy when dealing with rare or unseen terms.

Unlike previous models, our approach reduces the need for predefined segmentation rules, as GPT utilizes contextual understanding to assist in identifying sentence boundaries. Our method effectively handles OOV words, paraphrased sentences, and structural variations without requiring extensive corpus-specific tuning. Additionally, our framework remains adaptable and scalable across different languages and text genres, overcoming the dataset limitations of earlier models.

Our approach combines automated alignment with expert validation, ensuring both precision and adaptability, particularly LTC. By addressing key challenges in previous methods, it offers a more flexible and efficient solution for parallel sentence generation.

We identified several challenges that impacted the accuracy of the alignment process. One significant issue was inconsistency in sentence length and structure between English and Arabic. This inconsistency is primarily due to some translations being shortened, generally translated, or not matching the source text. For example, the English word “Yes” is translated into Arabic as “نعم، بالتأكيد,” which is longer because the Arabic version adds emphasis and certainty with “بالتأكيد” (certainly). This problem is more commonly observed in translating English words that are not lexicalized in Arabic. For example, the English sentence “Metabolism is a word frequently used and seldom understood by dieters” is translated into Arabic as “الأيض هي كلمة كثيرًا ما تستخدم، ونادرًا ما يفهمها الأشخاص الذين يتبعون النظام الغذائي.” The Arabic translation explicitly expands “dieters” to “الأشخاص الذين يتبعون النظام الغذائي” (people who follow a diet), because Arabic does not have one lexical item to express the meaning of the English term “dieters”. In general, these differences arise due to variations in grammar, cultural preferences, and contextual expression.

Another specific challenge arose from instances where students added acknowledgments or extra information at the beginning of the Arabic texts, which were not present in the English versions. This inclusion created additional content in the Arabic translations that had no corresponding segments in the English texts, complicating the creation of accurate translation pairs.

Furthermore, biases inherent in GPT influenced the alignment process and introduced several challenges that required manual interventions to maintain alignment quality and consistency. These challenges included issues such as incomplete sentence alignments, duplication, language switching within sentences, repeated Arabic text, and formatting errors.

Collectively, these issues highlighted the need for optimizing alignment algorithms and incorporating manual interventions to ensure the accuracy and reliability of the generated sentence pairs. To address these challenges, manual interventions were employed to resolve cases where automated methods struggled, such as identifying, correcting, or removing misaligned pairs caused by cultural or linguistic variations. Additionally, flagged outputs with formatting errors or alignment issues were meticulously reviewed and corrected to ensure consistency and quality. These combined efforts not only improved the consistency and accuracy of alignments but also ensured that the generated pairs adhered more closely to contextual and semantic equivalence. The success of these interventions was validated by the high MQM quality scores obtained during human evaluations.

To strengthen the justification for using GPT, we conducted a comparative analysis of the embedding models employed in this study—Mpnet, LaBSE, Distiluse, and MiniLM—against existing sentence alignment methodologies. In comparison, Chimoto & Bassett (2022) used fine-tuned LaBSE embeddings to align sentences in their dataset, achieving 53.3% alignment accuracy and over 85% precision at the same similarity threshold. While our study shows that pre-trained models like Mpnet and LaBSE perform well without fine-tuning, it is important to acknowledge the differences in datasets. Their study focused on Luhya-English, which introduces unique alignment challenges, whereas our dataset of English-Arabic pairs benefits from broader representation in pre-trained models. This comparison highlights the adaptability of embedding models in different contexts and demonstrates the potential of combining them with GPT for generating accurate parallel sentences in resource-rich domains.

Conclusions

This study presents a comprehensive framework for leveraging the SauLTC and LLMs to generate high-quality parallel English-Arabic sentence datasets, addressing a critical gap in TE. By integrating advanced AI tools, such as GPT, and embedding models, the proposed methodology enhances the efficiency and accuracy of parallel sentence generation. The results highlight the effectiveness of this approach, with LaBSE achieving the highest alignment consistency. These findings suggest that AI can play a valuable role in supporting translation pedagogy and practice.

The results demonstrated the framework’s ability to generate 15,845 sentence pairs, with LaBSE exhibiting the highest consistency and aligning 95% of the sentence pairs at the 0.7 threshold. Human evaluation further validated the dataset’s quality with an MQM score of 98%, emphasizing its contextual accuracy and reliability in addressing alignment challenges.

Future directions will focus on expanding the framework to encompass the entire SauLTC, incorporating additional annotations and metadata to enhance the dataset’s applicability across diverse domains. Furthermore, exploring advanced alignment techniques such as retrieval-augmented generation (RAG) can reinforce the contextual relevance and semantic accuracy of parallel sentence generation. Since automatic alignments and human evaluations present some challenges due to variations in linguistic interpretation and contextual nuances, future improvements should focus on fine-tuning embedding models to better capture language-specific variations. Additionally, developing context-aware scoring systems that integrate machine learning classifiers with human annotations will enhance AI evaluation metrics, ensuring more accurate and reliable assessments. By continuing to refine and expand this work, translation educators and researchers can equip students with the necessary skills to navigate the evolving landscape of AI-driven translation services, ultimately enriching the global exchange of knowledge and cultural understanding.

Supplemental Information

Supplemental Information 1 GPT3 5 sentences.

The code that segments the text into individual sentences using ChatGPT, allowing for more precise alignment and analysis.

Supplemental Information 2 ChatGPT3 5 Create Segments.

The code used to divide the text into smaller segments for efficient processing by ChatGPT.

Supplemental Information 3 Saudi Learner Translation Corpus.

English text and corresponding Arabic translation.

Additional Information and Declarations

Competing Interests

The authors declare that they have no competing interests.

Author Contributions

Moneerh Aleedy conceived and designed the experiments, performed the experiments, analyzed the data, performed the computation work, prepared figures and/or tables, authored or reviewed drafts of the article, and approved the final draft.

Fatma Alshihri conceived and designed the experiments, performed the experiments, authored or reviewed drafts of the article, and approved the final draft.

Souham Meshoul conceived and designed the experiments, performed the experiments, analyzed the data, authored or reviewed drafts of the article, and approved the final draft.

Maha Al-Harthi conceived and designed the experiments, performed the experiments, authored or reviewed drafts of the article, and approved the final draft.

Salwa Alramlawi conceived and designed the experiments, performed the experiments, authored or reviewed drafts of the article, and approved the final draft.

Badr Aldaihani conceived and designed the experiments, performed the experiments, authored or reviewed drafts of the article, and approved the final draft.

Hadil Shaiba conceived and designed the experiments, performed the experiments, authored or reviewed drafts of the article, and approved the final draft.

Eric Atwell conceived and designed the experiments, performed the experiments, authored or reviewed drafts of the article, and approved the final draft.

Data Availability

The following information was supplied regarding data availability:

The code and data are available in the Supplemental Files.

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
