# Peer review of "Designing AI-powered translation education tools: a framework for parallel sentence generation using SauLTC and LLMs"

_PeerJ Computer Science, doi:10.7717/peerj-cs.2788_

## Round 0.1 · original submission · Major Revisions

Dear Authors,

Thank you for the submission. The reviewers’ comments are now available. It is not suggested that your article be published in its current format. We do, however, advise you to revise the paper in light of the reviewers’ comments and concerns before resubmitting it. The followings should also be addressed:

1. Please pay special attention to the usage of abbreviations. Spell out the full term at its first mention, indicate its abbreviation in parenthesis and use the abbreviation from then on.
2. Equations 1 is part of the related sentence. Attention is needed for correct sentence formation.

Best wishes,

Reviewer 1 ·

Basic reporting

- The article “Designing AI-Powered Translation Education Tools: A Framework for Parallel Sentence Generation using SauLTC and LLMs” presents a timely and thought-provoking framework for integrating AI-powered tools into translation education. This approach emphasizes a structured methodology for building a didactic corpus aimed at improving translation teaching. The data appears robust, statistically sound, and well-controlled. While the paper makes a valuable contribution to our understanding of how AI-powered translation tools can be designed to support translation education, there are several areas where revisions are necessary to clarify and strengthen the overall argument and structure of the article.

Experimental design

- The article would benefit from a more detailed exploration of the language pairs studied, particularly the diversity within the Arabic language. Arabic is a language that consists of multiple varieties. Including a brief section on what constitutes Modern Standard Arabic (MSA) would provide crucial context, especially for non-specialist readers, e.g., unfamiliar with the division between oral and written Arabic.

- Some statements in the paper present an “enthusiastic tone” that needs further substantiation or be toned down. For instance, on page 2, the authors state that the corpus includes an "innovative annotation scheme that aims to enhance the learning experience of the students." While the idea of an original tool is indeed appealing, a clearer explanation of how this enhancement occurs would improve the section. For example, does the tool create an awareness of equivalence between English and Arabic, or focus on error correction and post-editing? Additionally, clarifying its application, whether for translation learners or in translator training, would aid reader comprehension.

- A similar enthusiastic tone appears on page 3, where the authors describe the framework as "comprehensive and accurate" without adequate detail. More clarification on what makes the framework “comprehensive” and “accurate” would be valuable. For example, is it comprehensive in terms of the range of translation challenges it addresses, or the types of language pairs included?

- The composition of the participants, comprising 366 students and 48 instructors, would be beneficial for potential replication studies. Details such as the students’ age, gender, years of study, and level of English, as well as the instructors’ age, gender, and experience, would be useful. If this information is unavailable, it could be noted as a limitation of the study.

- In describing the corpus's versatility for “a range of applications” on page 4, additional clarification on what these applications entail would be useful. Specifically, the sentence “This system aids in both translator training and research on translation quality across language combinations, enhancing the practicality and efficiency of the corpus for a range of applications” could briefly mention some specific applications to support the claim of practical versatility.

- On page 8, the authors acknowledge ChatGPT’s potential for context misinterpretation, which needs review and correction. Providing a specific example of this misinterpretation would illustrate the point more effectively for readers.

- The categorization of errors into minor, major, and critical lacks illustrative examples, which would improve reader understanding. Providing one example for each error type in both Arabic and English would be particularly useful from a linguistic point of view. Without examples, it is not possible to imagine the 3 proposed categories.

- A brief explanation for the choice of GPT-3.5 Turbo over GPT-4 (or 4o) would strengthen the methodology section. Reasons such as accessibility, task suitability, or specific advantages of GPT-3.5 for this framework would enhance clarity.

- Tokenization often varies by language and model specifications. It would be beneficial to address why the token count remains identical (1000) for both English and Arabic, given these differences. This explanation would provide a more robust understanding of the tokenization process in relation to language variables.

- The mention of “inconsistencies in sentence length” on page 12 would be clearer with an example illustrating this variation. This would enhance understanding of the issue’s potential impact on translation quality.

Validity of the findings

- The data appears robust, statistically sound, and well-controlled, with clear design to support translation education. 7. Some domains in the dataset appear to overlap (e.g., “business” versus “management”). An explanation of how these domains were defined and differentiated would help clarify potential implications for data interpretation.

- The term “very high accuracy” with a score of 0.67 on page 11 would benefit from comparison with benchmarks or related studies to contextualize this accuracy level.

Additional comments

- On page 5, the authors refer to "various genres" without specifying them. For better alignment with Table 1, it would be helpful to link to the genres mentioned there, to ensure readers understand which text types were included.

- The phrase “potential to revolutionize the learning process” on page 12 reiterates the enthusiastic tone at times in the paper, mentioned above, without adequate justification. Either tone down this language or support it with specific examples would improve the tone’s consistency.

- The article presents a valuable framework for integrating LLMs in translation education, and its originality is commendable. The argument is well-developed, and the authors effectively meet the goals stated in the introduction. While the conclusion successfully highlights the study's limitations and future directions, as detailed in this review, the paper would benefit, as mentioned above, from additional contextual information, explicit language examples to substantiate claims, and clearer connections to the broader implications for translation education.

·

Basic reporting

1. The in-text citations are inconsistent in formatting.
2. Some sentences are overly long and complex.
3. The abstract should concisely summarize the research problem, methodology, results, and implications.
4. The introduction should clearly state the research gap and how your work addresses it. Consider reorganizing it to improve the logical flow of ideas and ensure that each paragraph transitions smoothly to the next.
5. Define all acronyms upon first use and use them consistently thereafter.
6. Ensure that terms are used accurately and consistently throughout the paper.

Experimental design

1. The methodology lacks detailed descriptions of key steps. For example, the specific prompts used with ChatGPT are not provided, nor is there an explanation of how the model was instructed to align sentences.
2. The text mentions minimal preprocessing but does not elaborate on the techniques used to clean and prepare the data.
3. The process of segmenting the texts into smaller units is mentioned but not thoroughly explained.
4. Specific parameters used in ChatGPT (e.g., model version, temperature settings) and in the embedding models are not detailed.
5. The paper mentions the use of cosine similarity and human evaluation but does not specify the thresholds or criteria used to assess quality.
6. The rationale behind this sample size and selection process is not explained.
7. The paper does not compare the proposed method with existing sentence alignment techniques or tools, which could strengthen the justification for using ChatGPT.

Validity of the findings

1. The study focuses solely on health-related texts. This raises questions about whether the findings can be generalized to other domains within the SauLTC.
2. The impact of different text genres on the alignment process is not discussed.
3. The paper lacks detailed analysis, such as the distribution of scores, standard deviations, or how these scores correlate with alignment quality.
4. There is no discussion of what cosine similarity score is considered acceptable for a good alignment.
5. Human evaluation is inherently subjective. The paper should describe how evaluator bias was minimized and whether multiple evaluators were used to ensure reliability.
6. The human evaluation results are summarized but lack detailed findings, such as examples of errors found or common issues identified.
7. The paper briefly mentions challenges like misalignments but does not discuss how these were addressed or how they impact the overall findings.
8. Potential biases inherent in using ChatGPT, such as its handling of certain linguistic structures or idiomatic expressions, are not discussed.

Additional comments

1. The paper's structure could be improved for better flow.
2. The writing could be more concise and clearer. Avoid overly complex sentences that may confuse readers.
3. Ensure that the language used is formal and appropriate for an academic paper.
4. The conclusion should more effectively summarize the findings and their significance for translation education.
5. Provide specific suggestions for future work, such as expanding the framework to other domains or integrating additional LLMs.
6. Ensure all in-text citations are included in the reference list and that all references are correctly formatted.

---

## Round 0.2 · Minor Revisions

Dear Authors,

Thank you for submitting your revised article. Reviewers have now commented on your study. Although one reviewer accepts your article, it has still not been recommended for publication in its current form. However, we encourage you to address the concerns and criticisms of Reviewer 2 and to resubmit your article once you have updated it accordingly.

Best wishes,

Reviewer 1 ·

Basic reporting

Nothing else to add. Thank you.

Experimental design

Nothing else to add. Thank you.

Validity of the findings

Nothing else to add. Thank you.

Additional comments

I am happy with how the feedback provided has been taken into account by the authors. Thank you.

·

Basic reporting

1. Clearly define what is new in your approach beyond integrating existing tools like SauLTC, LLMs, and sentence embeddings. Justify why your framework is a significant advancement over previous methods.
2. Provide a more robust discussion on why this method is superior to other existing parallel sentence generation techniques.
3. Explain whether the proposed framework is applicable beyond Arabic-English parallel sentences and specify the limitations regarding other languages and domains.
4. Discuss the known biases, inconsistencies, and risks of using GPT for sentence alignment. Explain what measures you have taken to mitigate these issues.
5. Provide a clear explanation of how cosine similarity scores and human evaluation results are reconciled. Define how disagreements between automated and human assessments are handled.
6. Specify the qualifications of the evaluators, describe how they were trained, and report inter-rater reliability to validate the consistency of human judgments.
7. Explain why the study primarily focuses on health-related texts. Provide reasoning for the domain selection and discuss how findings apply to other fields within SauLTC.
8. Address whether the dataset is large enough for AI-powered applications. Explain how the framework could scale to accommodate larger datasets.
9. Streamline the related work section to highlight key studies that are directly relevant to your research. Identify gaps and clearly position your study within existing literature.
10. Avoid repeating the same content across sections. Ensure each section contributes new insights rather than reiterating previous statements.
11. Ensure all citations follow a consistent format. Verify that every reference is properly integrated into the text and that all sources are accurately cited.
12. Revise sentences for clarity, conciseness, and grammatical accuracy. Simplify unnecessarily complex structures to improve readability.
13. Define key terms such as "parallel corpus," "parallel sentence generation," and "alignment," and use them consistently throughout the paper.
14. Acknowledge potential drawbacks of the framework, including ethical concerns, biases in AI-generated translations, and practical challenges in implementation.
15. Refrain from making claims that overstate the precision and contextual appropriateness of AI-generated translations. Provide evidence to support any strong assertions.
16. Clearly outline concrete steps for future research, such as dataset expansion, improved AI evaluation metrics, or alternative alignment techniques.

Experimental design

no comment

Validity of the findings

no comment

Additional comments

no comment

---

## Round 0.3 · accepted · Accept

Dear Authors,

Thank you for clearly addressing the reviewers' comments. Your manuscript now seems ready for publication.

Best wishes,

·

Basic reporting

-

Experimental design

-

Validity of the findings

-

Additional comments

-